# Role of Mesenchymal Stem/Stromal Cells (MSCs) and MSC-Derived Extracellular Vesicles (EVs) in Prevention of Telomere Length Shortening, Cellular Senescence, and Accelerated Biological Aging

**DOI:** 10.3390/bioengineering11060524

**Published:** 2024-05-21

**Authors:** Myrna Y. Gonzalez Arellano, Matthew VanHeest, Sravya Emmadi, Amal Abdul-Hafez, Sherif Abdelfattah Ibrahim, Ranga P. Thiruvenkataramani, Rasha S. Teleb, Hady Omar, Tulasi Kesaraju, Tarek Mohamed, Burra V. Madhukar, Said A. Omar

**Affiliations:** 1Division of Neonatology, Department of Pediatrics and Human Development, College of Human Medicine, Michigan State University, East Lansing, MI 48824, USA; gonza990@msu.edu (M.Y.G.A.); abdulhaf@msu.edu (A.A.-H.); ibrahi22@msu.edu (S.A.I.); thiruve5@msu.edu (R.P.T.); rasha.saifel-dein@med.svu.edu.eg (R.S.T.); omarhady@msu.edu (H.O.); kesaraj1@msu.edu (T.K.); mohame54@msu.edu (T.M.); madhukar@msu.edu (B.V.M.); 2College of Human Medicine, Michigan State University, East Lansing, MI 48824, USA; vanhee12@msu.edu (M.V.); emmadisr@msu.edu (S.E.); 3Regional Neonatal Intensive Care Unit, Sparrow Hospital, Lansing, MI 48912, USA; 4Histology and Cell Biology Department, Faculty of Medicine, Mansoura University, Mansoura 35516, Egypt; 5Department of Pediatrics and Neonatology, Qena Faculty of Medicine, South Valley University, Qena 83523, Egypt

**Keywords:** telomere length, biological age, mesenchymal stem cells, extracellular vesicles, exosomes

## Abstract

Biological aging is defined as a progressive decline in tissue function that eventually results in cell death. Accelerated biologic aging results when the telomere length is shortened prematurely secondary to damage from biological or environmental stressors, leading to a defective reparative mechanism. Stem cells therapy may have a potential role in influencing (counteract/ameliorate) biological aging and maintaining the function of the organism. Mesenchymal stem cells, also called mesenchymal stromal cells (MSCs) are multipotent stem cells of mesodermal origin that can differentiate into other types of cells, such as adipocytes, chondrocytes, and osteocytes. MSCs influence resident cells through the secretion of paracrine bioactive components such as cytokines and extracellular vesicles (EVs). This review examines the changes in telomere length, cellular senescence, and normal biological age, as well as the factors contributing to telomere shortening and accelerated biological aging. The role of MSCs—especially those derived from gestational tissues—in prevention of telomere shortening (TS) and accelerated biological aging is explored. In addition, the strategies to prevent MSC senescence and improve the antiaging therapeutic application of MSCs and MSC-derived EVs in influencing telomere length and cellular senescence are reviewed.

## 1. Introduction

Aging is a biological process defined as a progressive functional decline in tissue and organs that eventually results in death [1]. The functional decline can result from the loss or diminished function of post-mitotic cells or from failure to replace such cells due to inability of stem cells to maintain replication and cell divisions. Aging is a natural process, and the biology of aging varies among individuals [2]. López-Otín et al. proposed twelve hallmarks of aging that are interconnected with each other. These are genomic instability, telomere attrition, epigenetic alterations, loss of proteostasis, disabled macroautophagy, deregulated nutrient-sensing, mitochondrial dysfunction, cellular senescence, stem cell exhaustion, altered intercellular communication, chronic inflammation, and dysbiosis [3].

### 1.1. Telomere Length, Cellular Senescence, and Normal Biological Age

Telomeres are specific DNA protein structures that represent repetitive sequences of DNA (TTAGGG) and associated protective six-protein complexes known as shelterin [1]. TRF1 and TRF2 are two shelterin proteins that bind double-stranded telomeric DNA, while the POT1-TPP1 shelterin coats and protects the single-stranded telomeric G-rich overhang that is characteristic of all chromosomal ends [4]. Telomeres serve to stabilize the chromosomes and protect them from degradation. Telomeres also serve as biomarkers of oxidative stress and inflammation [5]. The telomere length (TL) progressively shortens with each cell division, and once it reaches a critical level, the cell will either stop proliferation and enter a state of senescence or undergo apoptosis [4].

Cellular senescence was first described in the 1960s by Hayflick and Moorhead [6]. Cellular senescence refers to an irreversible cell cycle arrest associated with changes in cell DNA, morphology, secretory profile, and limited replicative potential [7]. These changes can lead to various types of accelerated cellular senescence. The first type is replicative senescence induced by DNA damage or telomere shortening (TS). The second type is independent of TS, and it is termed stress-induced premature senescence (SIPS), which is caused by high oxidative stress. Additionally, cellular senescence could occur due to oncogene activation or oncogene-induced senescence [8]. 

Normal biological aging depends on many factors. Tissue biological aging can be caused by cells progressively undergoing senescence and losing their ability to proliferate, which is essential for replacing damaged cells [6]. Another factor that affects biological aging is cell fate. Cell fate depends on the intensity and duration of the initial stimulus, as well as the cell type [9]. Replicative senescence occurs in response to nuclear DNA damage, which activates a signaling cascade defined as DNA damage response (DDR). This cascade ultimately leads to p53 activation, which in turn elicits cell cycle arrest. Prolonged DDR activation triggers senescence. Telomere theory states that short and dysfunctional telomeres cannot be repaired by any of the known DNA repair mechanisms and consequently trigger a persistent DDR, which leads a cell to senescence [10]. The hallmarks of senescence include the arrest of cellular proliferation; and macromolecular damage that leads to a change in cell morphology, especially an increase in its size, deregulated metabolism due to mitochondrial dysfunction, and senescence-associated secretory phenotypes (SASPs). These SASPs include proinflammatory cytokines, chemokines, angiogenic factors, growth factors, prooxidants, proteases, and other secretory factors [11,12].

### 1.2. Accelerated Biological Aging

Accelerated biological aging results when the TL is shortened prematurely, and this can occur secondary to cellular damage from biological or environmental stressors, leading to a defective reparative mechanism, as summarized in Figure 1, panel B. While the specific roles of intrinsic and extrinsic molecular factors are unclear, the damages induced to tissue and responding tissue-specific stem cell (TSC) proliferation are suggested to promote accelerated TSC telomere shortening through continued cycles of damage and TSC differentiation-driven repair. This suggestion is further elaborated in a study investigating this process in the airway, conducted with the epithelial tissue of mice and humans exposed to naphthalene, which concluded the association of repeated injury with shorter tracheobronchial TSC telomeres and a shorter lifespan [13].

The progressive loss of TL with multiple mitotic cell division is associated with cellular senescence, but the progressive loss of TL in human somatic cells is believed to act as a tumor-suppressor mechanism that limits clonal proliferation, prevents clonal dominance, and ensures a polyclonal composition of these cells [14]. While TS may act as a tumor-suppressor mechanism, it may also promote tumor growth by driving the selection of cells with defective DNA damage [15].

More specifically, panel B summarizes the proposed mechanism for MSCs and their derived EVs in influencing cellular senescence and biological aging. MSCs are a type of TSC that, through paracrine action, including EVs, could influence this process by delaying normal aging or preventing accelerated premature biological aging. EVs play a role in this process through different mechanisms that include the transfer of mRNA, proteins, or direct transfer of telomeric DNA between cells. In both scenarios, TS can lead to cellular senescence or apoptosis. Cellular senescence resulting from telomere attrition is one of the fundamental contributors to the overall process of organismal aging.

### 1.3. Telomerase Role in Maintenance of TL

Telomerase is a ribonucleoprotein enzyme that synthesizes telomeric DNA to counter TS and is the main pathway to rescue and reverse TL and is expressed in stem cells, germ cells, and a few immune cells after birth [4]. Telomerase reverse transcriptase (TERT) is a key subunit of the telomerase complex and critical to telomere preservation, with its primary role being the maintenance of TL through the addition of TTAGGG repeats. This serves to counter the degradation of chromosomal ends over the course of continued replication [16].

Both inherited and acquired mutations in the TERT gene lead to premature TS and cellular senescence. Dyskeratosis congenita is a common example used to describe diseases associated with TS, and it is due to genetically inherited mutations of the TERT gene [17]. However, TERT is also involved in extratelomeric roles, including effects on proliferation, senescence, differentiation [18], and apoptosis [19] of cells. For example, TERT was found to play a protective role in preserving proliferative capacity in alveolar type II pneumocytes and knockout of TERT gene increased senescence and enhanced inflammation and fibrosis in response to epithelial injury. This effect was not induced through TS [16].

## 2. Factors Contributing to TS and Accelerated Biological Aging

Multiple factors, such as inflammation and oxidative stress, can lead to TS. This section explores how those factors can contribute to TS shortening and the potential effects on biological aging.

Repeated inflammation accelerates biological aging through rapid leukocyte proliferation and accumulation of pro-inflammatory cytokines, including IL-6 and TNF-alpha, and has been associated with short leukocyte telomeres in adults [20]. Upon sufficient TS to induce cellular senescence, a positive feedback loop is created as these cells adopt SASP, which is characterized by increased nuclear factor kappa-beta (NF-kB) activity and overexpression of IL-6, TNF-alpha, and IFN-gamma in circulating macrophages [21]. Telomere damage from such settings as inflammation from infection or other stress effectors also promotes mitochondrial dysfunction, reducing its capability to reduce ROS. A bidirectional communication loop has been identified between the mitochondria and nucleus, mediated by mitochondrial translocation of hTERT from the nucleus, where it binds mitochondrial DNA (mtDNA) and mitochondrial tRNA [22]. Communication back to the nucleus is suggested to indicate the oxidative state of the mitochondria through the translocation of the telomerase RNA component of hTERC [23]. In the setting of telomere damage, subsequent p53 tumor-suppressor gene activation suppresses activity of PGC1-alpha and PGC1-beta promoters, which serve as significant regulators of mitochondrial metabolism [24]. Impaired mitochondrial physiological function induces a positive feedback loop as less ROS is eliminated by the mitochondria, and oxidative stress damage to the telomeres continues to accumulate [25]. Being critical to the process of telomere maintenance, telomerase represents another facet of the interaction of biological aging and inflammation. Its function is critical to preventing apoptosis or senescence, and its activity decreases as cells progressively succumb to premature senescence from ROS [25].

Oxidative stress presents a well-characterized contributor to increased biological aging. Oxidative stress results from a loss of balance between ROS and antioxidant protective systems. Oxidative stress accelerates telomere loss, whereas antioxidants decelerate it. Cellular replicative senescence is heterogeneous even within a clonal-derived cell population growing under low-stress conditions, and the rates of cellular aging and TS depend on the balance between oxidative stress and antioxidative defense [26].

Telomeres are at a higher risk of nucleic acid damage due to the high guanine content within the telomeric repeats, as these can be more prone to oxidative damage [27]. This effect extends to prenatal exposure. Recent studies have shown that maternal oxidative stress from smoking and maternal nutrition (obesity) results in shorter telomeres [28]. Several studies have modeled oxidative stress through hyperoxia treatment, showing TS in human fibroblasts [29]. The influence of oxidative stress in aging is further supported by the deceleration of TS by antioxidants [30].

## 3. Role of Stem Cells in TS, Cellular Senescence, and Advanced Biological Age

One of the current evolving branches in science is stem cell research for the potential role of stem cells in tissue homeostasis, aging-related diseases, and cellular therapy.

The main function of stem cells is maintaining tissue homeostasis through renewing the impaired cells. The overall decrease in the regenerative ability of tissues is one of the most common characteristics of aging. When senescence occurs in stem cells, the potential of tissue regeneration will decrease, which contributes to aging [31]. Senescent cells can drive stem cells to re-enter the cell cycle through SASP, which accelerates the exhaustion of stem cells [1]. Those mediators can also disturb the specialized microenvironment, or niche, which is essential for the optimal function of stem cells [32]. In addition, proteases secreted by senescent cells may induce the cleavage of extracellular matrix proteins or other components in the tissue microenvironment, and these changes eventually lead to the impairment of tissue homeostasis, thus leading to tissue aging [33,34]. Liu et al. recently reviewed the characteristics of tissue-specific stem cells during aging and the factors that control their self-renewal and differentiation to replenish their pool. The cycles of tissue injury and repair drive stem cell proliferation and rapid replication. As this depletes the stem cells’ reparative capacity, individuals may have varying onsets of repair, affecting the severity of the disease process and thus increasing morbidity and mortality [35].

Individuals with different initial TLs in their stem cell pools could show varying disease presentations due to the variable number of divisions allowed before telomeres reach a critical length, resulting in senescence and abnormal tissue repair. The generation of oxidizing agents by inflammation and their effect on stem cells was shown in a study observing the immune privilege of rat stem cells in bone marrow transplants, and the researchers observed that inflammation resulted in the oxidation of membrane phosphatidylcholines that created inflammation, promoting damage-associated molecular patterns that initiated the cellular rejection process [36].

The resulting aging effects and oxidative stress from chronic inflammation are thought to be the effects of the increasing expression of NF-κβ [37]. A possible mechanism for these aging effects includes bone fragility, skin dystrophy, muscle weakness, and reduced neurogenesis. These factors may be due to the loss of stem cell proliferative capacity from the inflammatory effects of a higher expression of NF-κβ. Its expression drives the inflammatory response, resulting in increased cytokine expression, increased lymphocyte activation, and increased generation of ROS [20,21,26,29,36,38]. These factors most likely accelerate the telomere attrition of stem cells and result in cellular senescence and accelerated biological aging.

The shortening of telomeres occurs with each cell division; therefore, rapid replication in the setting of repeated cycles of tissue repair drives a significantly accelerated erosion of telomeres. The progressive damage and repair in tracheobronchial tissue results in tracheobronchial-specific stem cells showing reduced TL and premature cellular senescence [13]. In a recent study, lung injury was induced by administering naphthalene to mice lungs and cultured human tissue to observe this aging influence of tracheobronchial stem cells [13]. Of the activated stem cell pool, ninety-six percent underwent terminal differentiation, with short-lived stem cells possessing significantly shorter telomeres [13]. The repeated naphthalene exposure increased the biological age of the stem cell population and depleted the number of cells. This combination was suggested to be a mechanism for aberrant repair and development of chronic lung disease [13].

It has been suggested that continued injury and repair in chronic wounds result in compromised progenitor and stem cell function [39]. This reduced function, as well as the depletion of stem cell pools observed in the frequently cycling tissue of chronic epidermal wounds [40], could be induced by accelerated telomere erosion, resulting in this premature stem cell aging.

## 4. Mesenchymal/Stromal Stem Cells (MSCs)

Mesenchymal stem cells, also called mesenchymal stromal cells (MSCs), are multipotent stem cells of mesodermal origin. MSCs are examples of TSCs that can differentiate into other types of cells with specific growth and differentiating local factors in specific organs. For example, they can differentiate into adipocytes, chondrocytes, and osteocytes [41]. MSCs can be isolated from a variety of adult and fetal tissues and are most often isolated from BM stromal cells from the tibia, fibula, iliac crest, or the thoracic and lumbar spine. They also can be isolated from other sources, such as adipose tissue; skeletal muscle; deciduous teeth; synovium; and gestational tissues, such as Wharton’s jelly, umbilical cord vein endothelial cells (UCVECs), and the placenta [42,43,44]. Human bone marrow-derived mesenchymal stromal cells (BM-MSCs) have been widely studied and used for treating multiple degenerative disease conditions and tissue repair [45]. BM-MSCs can be readily isolated, but there are some disadvantages, including painful invasive isolation and limited cell numbers [46].

MSCs are of interest for their ability to modulate the immune system and their potential application in tissue regeneration and engineering in numerous human diseases, such as osteoporosis, diabetes mellitus, and retinal degeneration [47]. The survival of engineered cells through genetic alteration can lead to optimized functional cells and minimized side effects of disease processes. These genetic alterations can ultimately alleviate aging-related diseases. However, the translation of MSCs-based therapies has been hindered by the heterogeneity of the isolated cells and the lack of standardized methods for their definition and characterization, especially the absence of specific surface markers. The International Society for Cellular Therapy (ISCT) proposed minimal criteria for the definition of human MSCs to increase consistency in their verification. Three criteria to define MSCs were proposed, including adherence to plastic in standard culture conditions; phenotype positive (>95%) for CD105, CD73, and CD90 and phenotype negative (<2%) for CD45, CD34, CD14 or CD11b, CD79a, CD19, and HLA-DR; and in vitro differentiation of osteoblasts, adipocytes, and chondroblasts (demonstrated by staining of in vitro cell culture) [48]. Kundrotas et al. showed that early passage/young MSCs are genomically stable and maintain their identity and high proliferative capacity. In later passages, despite the chromosomal stability, old MSCs are more senescent and show a slower proliferation capacity, with altered morphology and immunophenotype [49]. So, in this review, we primarily focus on MSCs derived from gestational tissues as a possible alternative source of MSCs with high proliferative capacity and less tendency for cellular senescence. Among all the sources of MSCs, such as adipose tissue, bone marrow, and peripheral blood, those derived from the gestational tissues (umbilical cord and placenta) have the advantage of being a feasible unlimited source of stem cells. Moreover, these tissues are considered medical waste and cause no harm to the donor and pose no ethical dilemmas. In comparison with adult MSCs, the MSCs of fetal origin show higher plasticity, more rapid proliferation [50], lower immunogenicity [51], lower risk of infection, and the possible higher stemness potential related to the early embryological origin of the placenta [52]. Fetal MSCs were reported to have a transcription factor profile like that of embryonic stem cells (OCT4, SOX2, and NANOG), without the risk of teratoma formation reported with embryonic stem cells [53]. Other studies have also reported that there is no significant change in the number of cells and growth kinetics between BM MSCs and placenta MSCs [54]. In our neonatal immunology and stem cell research laboratory, we found that cultured cells from the placental tissues of healthy full-term pregnancies exhibit MSCs’ morphological features: they have adherent and spindle-shaped fibroblast-like cells and have the flow cytometric characteristics of MSCs [55]. Placental stem cells were reported to have unique immunomodulatory and immune-suppressive characteristics [56]. MSCs have been isolated from several placental regions, namely (i) from the fetal membranes, which yield human amniotic mesenchymal stromal cells (hAMSCs), and human chorionic mesenchymal stromal cells (hCMSCs) [57]; (ii) from the chorionic villous stroma of first-trimester placenta [56]; (iii) from at least five compartments of the umbilical cord, namely the umbilical cord blood, the umbilical vein sub-endothelium, and three regions of the Wharton’s jelly (i.e., the perivascular zone, the intervascular zone, and the subamnion) [58]; (iv) from different regions of the decidua [59]; (v) from the fetal chorionic villi of term placenta [60].

## 5. Cellular Senescence and Functional Decline of MSCs with Aging

The concept of cellular senescence can also be applied to MSCs. As MSCs age, they exhibit distinctive phenotypic alterations. It is important to understand the factors that affect MSCS and lead to their senescence and their premature aging. When MSCs are aging, their dynamics and immunoregulatory activities are altered, resulting in impaired therapeutic potential [61,62]. The in vivo administration of senescent MSCs is associated with systemic inflammation and the impairment of proliferation and differentiation [63,64]. As MSCs age, they exhibit distinctive phenotypic alterations that include the enlargement of their size and flattening of their shape morphologically. Senescent MSCs include cell cycle arrest; the appearance of SASP; a flat hypertrophic shape; and high activity of senescence-associated β-galactosidase (SA-β-gal), which is the gold standard and the most widely accepted marker for evaluating cellular senescence [65].

Below is a summary of the possible mechanisms of MSC cellular senescence, as reviewed by Weng et al. [66]. Oxidative stress is commonly connected with inflammation in premature senescence due to its role in developing pro-inflammatory SASP, as previously discussed [35]. ROS promotes MSCs’ senescence by causing oxidation and a cascade of secondary metabolic reactive species. The mitogen-activated protein kinase p38 (p38 MAPK) is known to be an indispensable molecule in mediating ROS-induced senescence through promoting p53 phosphorylation, predisposing to senescence through the p53/p21^CIP1/WAF1^ pathway [67,68]. The phosphatidylinositol-3-kinase (PI3K)-protein kinase B (AKT) signaling pathway is also involved in ROS-activated MSCs’ senescence through promoting the transcription of its target genes, including mammalian target of rapamycin (mTOR), forkhead box protein O3 (FOXO3), and p53 [65]. The production of ROS and ROS-derived damage may be a major player throughout the MSCs’ aging process, especially the casual link between mitochondrial dysfunction and ROS [69].

Another factor that could induce senescence in MSCs is the accumulation of DNA damage that occurs in the form of DNA-replication errors, telomere attrition, or both. Multiple factors affect this process, such as oncogene activation, irradiation, chronic inflammation, and oxidative stress. DNA damage response (DDR), one of the hallmarks of MSC senescence, is a dominant regulator of cell cycle arrest. This process is primarily regulated by the cyclin-dependent kinase (CDK) inhibitors p21^CIP1/WAF1^ and p16^INK4A^ signaling pathways, which antagonize CDK to block cell cycle progression [70]. Most of the p16^INK4A^-positive cells are senescence-associated SA-β-gal-positive, and the knockdown of the p16^INK4A^ gene in senescent human MSCs reduced the number of senescent cells and restored their ability to proliferate [71]. The MSCs’ epigenetic alteration implicated in the MSCs aging includes histone modification, chromatin remodeling, and DNA methylation. These events may drive MSCs’ senescence-related manifestations. In aging MSCs, histone deacetylase (HDAC) deficiency upregulates lysine-specific demethylase 6B or KDM6B and downregulates polycomb genes through the RB/E2F pathway, working as a transcriptional activator of p16^INK4A^ by demethylating H3K27me3 and leading to senescence [72].

The stem cell niche is a unique microenvironment surrounding stem cells that has been shown to support, regulate, and maintain stem cells in a quiescent state, preventing genetic damage [73]. SASP plays a pivotal role in inflammatory change in an autocrine and paracrine manner. Adipocytes, one type of niche-supporting cells, have been found to accumulate in the bone marrow niche, along with aging, and have been postulated as the primary source of proinflammatory cytokines, resulting in a pro-inflammation milieu for MSCs [74,75]. Zhang et al. indicated that the Wnt/β-catenin signaling may play a critical role in MSC aging induced by the serum of aged animals and suggested that the DNA-damage response and p53/p21 pathway may be the main mediators of MSC aging induced by excessive activation of Wnt/β-catenin signaling [76]. Moreover, mice of different ages, when surgically conjoined, resulting in a shared blood circulation, demonstrate that exposure to young circulation improves the osteoblastic differentiation capacity of aged MSCs in vivo and that β-catenin can reverse the diminished fracture repair phenotype of the older mice [77]. This rejuvenation effect, or antiaging effect, suggests the existence of some detrimental substances in aged circulation, limiting MSCs’ functionality and resulting in cellular senescence. The increased β-catenin levels have been identified as one of the harmful circulating factors, and blocking the canonical WNT pathway could ameliorate the effects caused by aging serum [76,77]. Hyperglycemia is a prominent feature of diabetic patients, has been shown to alter MSCs’ characteristics and functions, and results in MSC senescence. Special attention should be paid when considering autologous MSC therapy in T2D, as its therapeutic potential may be restricted [78,79]. Modification with genetic material, noncoding RNA and exosomes, intracellular signaling pathways, impaired differentiation, reduced capacity for self-renewal, and proliferation play a critical role in MSCs’ senescence and functional decline [80]. The spontaneous malignant transformation of murine MSCs after a long-term in vitro culture has been described, and a similar transformation of human MSCs has also been investigated. It was reported that human MSCs immortalize at high frequency in long-term cultures and undergo spontaneous malignant transformation. Furthermore, many of these studies have noted that transformed cells occur in the human MSCs cultures due to the cross-contamination of the original cell culture with tumor cells [81]. Therefore, there is still ambiguity as to whether murine MSCs or human MSCs also have a higher tendency to be transformed into a malignant cell type after in vitro culturing [81,82]. It has also been found that aged MSCs cannot be used for transplantation and that the transplantation of aged MSCs into elderly patients is less effective [83]. Thus, the assessment of MSCs for the percentage of aneuploidy cells before using them in clinical applications is recommended to decide how to combat MSCs’ senescence.

## 6. Role of Gestational Tissue-Derived MSCs in the Prevention of TS, Cellular Senescence, and Accelerated Biological Aging

MSCs have a stable TL compared with differentiated cell lines. However, MSCs still have low levels of telomerase; therefore, telomeres slowly shorten [84]. Current ongoing research proposes protection of telomere or increasing telomerase activity to reduce cell aging and consequently prevent cellular senescence. The introduction of hTERT into MSCs resulted in a substantial multiplication of their replicative lifespan accompanied by the preservation of normal karyotype, elongation of telomeres and loss of the senescent phenotype without impact on differentiation ability [85,86]. Trachana et al. [87] examined the genetic modification of cells with hTERT and the effects of oxidative damage in adult stem cells from adipose tissues as well as stem cells from umbilical cord Wharton jelly and showed that those cells had an enhanced growth potential which can improve the antioxidant capabilities. This study thus shows the importance of telomerase activity. A supporting study by Yamashita [88] examined SIRT1, which increases hTERT transcription, preventing replicative senescence in HUC-F2 cells. Chen et al. [89] examined MSCs and SIRT1 effects in preventing age-related cellular senescence, where overexpression of SIRT1 induced TERT and leads to increased cell proliferation. The study also examined TL in aged MSCs, which showed a decrease in SIRT1 expression and supported the possibility that SIRT1 aids with telomerase activity and TL. This could also be through the expression of shelterin proteins such as TPP1, whose expression could be upregulated by SIRT1.

Placental-derived MSCs may also play a vital role in preventing accelerated biological aging and TS. Previous studies have shown that MSCs from amniotic fluid (AF-MSCs), amnion membrane (AM-MSCs), and endometrium (EM-MSCs), as well as Wharton’s jelly MSCs, have relative epigenetics stability, which suggests that amniotic fluid-derived MSCs could be a favorable type of MSCs for cellular therapy [90] The exhaustion of functional stem cells is critical in aging and aging-associated degenerative diseases. Stem cell transplantation is generally a promising candidate method for regenerative applications because stem cells possess a high proliferative capacity and can differentiate into other cell types. Pipes et al. [91] studied the replicative potential after transplant from umbilical cord blood vs. PBHSC transplant. The study showed that there was a longer TL in patients who received Umbilical Cord Blood (UCB) transplants. Moreover, in a premature-aging mice model of Bmi-1 deficiency, transplanted amniotic membrane MSCs (AMSCs) inhibited oxidative stress and DNA damage in multiple organs. The transplanted AMSCs carried Bmi-1 migrated into multiple organs, proliferated and differentiated into multiple tissue, promoted growth, and improved the premature senescent in Bmi-1-deficient transplant recipients [92]. These findings indicate that AMSC transplantation ameliorated the premature senescent phenotype and could be a novel therapy to delay aging and prevent aging-associated degenerative diseases. When BM-MSCs were compared with adipose stem cells (ASCs), as well as umbilical MSCs, they became senescent in earlier passages of subculturing, and there was also a decrease in TL with increased passages than those of ASCs and UCMSCs [93]. Therefore, there may be a greater benefit to using cells from the gestational tissues of umbilical cord blood or the placenta, as they have cellular senescence in later passages compared to BM-MSCs. Umbilical-cord MSCs, compared with BM, have a faster doubling rate and can maintain TL through a more replicative cycle.

## 7. Role of Aged MSCs in Pathogenesis of Diseases

Increasing evidence has shown that senescent MSCs play degenerative roles during tissue repair and may contribute to the pathogenesis of age-associated diseases.

MSCs derived from osteoarthritis (OA)-affected cartilage were found to express higher levels of hypertrophic cartilage markers, which were further increased with the induction of chondrogenesis. This suggests that OA-MSCs may contribute to the pathogenesis of this disease [94]. Other physical-disability disorders were also connected to MSC senescence. Transplanting senescent adipose-derived MSCs into young mice caused those mice to exhibit early walking disability and accelerated aging [95]. Senescent MSCs were also investigated as the underlying pathology in interstitial pulmonary fibrosis (IPF). An increase in cell-senescence markers was found in lung fibroblasts from IPF patients. MSCs isolated from IPF patients showed an increase in cell senescence, which was associated with increased DNA damage and SASPs [96]. The study of cells isolated from patients with cardiovascular disease showed an accumulation of senescent cardiac progenitor and stem cells (CPCs) with diminished self-renewal, differentiation, and regenerative potentials. Those cells showed an increased expression of SASPs, including IL-1β, IL-6, IL-8, MMP-3, PAI1, and GM-CSF [97,98].

## 8. Strategies to Prevent MSCs’ Senescence and Improve Their Clinical Application

There is a need to develop strategies for MSCs expansion to produce a large number of cells with retained stemness and lineage plasticity. Multiple approaches have been tried to prevent or reverse MSCs’ senescence to improve their clinical application. Antioxidants such as ascorbic acid can inhibit ROS production in MSCs through AKT/mTOR signaling [99], and senescence suppressors such as SIRT1 may attribute to the maintenance of genomic stability and metabolic efficiency. Depletion of SIRT3 accelerated aging and inhibited MSCs’ differentiation into osteoblasts and adipocytes, and the overexpression of SIRT3 in late-passage MSCs could restore their differentiation capacity and reduce oxidative stress and senescence [100]. Genetic approaches, such as genetic engineering, have been proposed to slow down MSCs’ senescence effectively. It has been reported that the knockdown of the migration inhibitory factor (MIF) could induce senescence in young MSCs, while MIF overexpression leads to the rejuvenation of aged MSCs [101]. The overexpression of hTERT can increase the replicative life span of MSCs, preserve normal karyotype, promote telomere elongation, and abolish cellular senescence [85]. Moreover, miR-195 knockdown could activate telomere re-lengthening by upregulating the expression levels of hTERT and reverse the senescence clock in aged stem cells by telomerase, contributing to the rejuvenation of MSCs’ senescence [102]. A rejuvenation effect has been observed in aged human MSCs associated with the reversal of their senescent phenotype and an enhancement of their resistance to oxidative stress by the overexpression of Erb-B2 receptor tyrosine kinase 4 [103]. Ahmed et al. [22] examined the protective role of TERT overexpression in human MRC5 fibroblasts, as assessed through the measurement of mitochondrial superoxide levels and mitochondrial membrane potential in the setting of hyperoxia. They found reduced superoxide levels and elevated membrane potential with TERT overexpression, along with increased TERT mitochondrial protein, as measured by a TERT Western blot analysis, and increased telomerase activity, as measured by the telomere repeat amplification protocol. Multiple factors have the potential for reversing stem cells’ aging and restoring the regenerative function of the aged stem cells. For example, the upregulation of SIRT6 expression is proposed to control aging by potentially silencing repetitive elements and achieving epigenetic rejuvenation [104]. The development of gene-editing technologies such as CRISPR-Cas9 systems may enhance the available genetic strategies to engineer stem cells to overexpress or suppress gene expression. Stem cells therapy is a potential way to supplement stem cells to attenuate aging. Exposure to a young systemic environment can be associated with stem cell revitalization and the systemic rejuvenation of aged tissues [105,106,107].

Senolysis is the process of removing senescent cells from proliferative cells by using specific drugs that selectively clear apoptosis-resistant senescent cells. The use of senolytic drugs is designed to target senescent cells and alleviate the symptoms of multiple aging phenotypes [108,109]. For example, A β-galactosidase-targeted prodrug has been proven to have the ability to clear senescent cells, attenuate low-grade local and systemic inflammation, and eventually restore the physical function of different tissues [109].

The enhancement of the knowledge of the biological, physiological, pathological, and molecular basis of stem cell aging may provide intriguing insights into potential therapeutic interventions for stem cells’ senescence and aging-related disorders [108].

## 9. Antiaging Therapeutic Application of MSCs

MSCs have been used in cell therapy, although their clinical usefulness remains restricted. MSCs of various donors are heterogeneous in vitro cell passages, and culture conditions directly affect the cell phenotype. The ability of MSCs to induce immune tolerance and their role in tissue regeneration has been investigated by many clinical studies. They also have been used in the treatment of many conditions, such as the induction of immune tolerance following solid organ transplantation [110], treatment of autoimmune disease, myocardial infarction, stroke, liver cirrhosis, ulcer healing, epidermolysis bullosa, and human herpes virus infection; they have also been associated with the acceleration of wound healing [66]. Clinical trials were conducted using placenta-derived stem cells for type 2 diabetes, Crohn’s disease [111], multiple sclerosis [112], pulmonary sarcoidosis [113], ischemic stroke and the treatment of hypoxic-ischemic encephalopathy [114]. Successful results have been reported for the treatment of neurological disorders such as Parkinson’s disease and spinal cord injury and in the treatment of major end organ fibrosis in animal models [115].

It is highly desirable to expand MSCs for multiple passages, with the goal of not having signs of senescence. MSCs from fetal and gestational tissues have more plasticity and grow faster than MSCs from bone marrow. These MSCs are better able to cope with the genotoxic stress that may occur either during in vitro cultivation or following transplantation in patients and may represent a valid alternative to BM-MSCs in regenerative medicine and be of great relevance. When MSCs were cocultured with various cell types, they improved the hallmarks of aging, including TL and cellular senescence. Different pathways have been involved in this effect. One example of the potential pathway is shown by coculturing Ad-MSCs with lung fibroblast cells that were induced for senescence. After the coculturing with the MSCs, the fibroblasts restored their proliferative capacity with a reduction of TS. This work also showed that the addition of MSCs was associated with the modulation of several factors related to cellular aging and cell senescence. In the same study, the authors showed that in vivo injection of these MSCs was associated with a reduction in the signs of aging [116]. Similar effects by BM-MSCs on doxorubicin-induced senescence in cardiomyocytes in vitro culture were also reported by Meng Hou et al. [117]. The inhibition of microRNA-34a may lead to the induction of SIRT1 expression, with subsequent attenuation of senescence and the improvement in TL reduction as the mechanism of action of BM-MSCs [117].

## 10. Paracrine Effect of MSCs by Extracellular Vesicles (EVs)

EVs are heterogeneous populations of nano- and micro-sized membrane-enclosed fragments of cytoplasm and bioactive materials (mRNA, miRNA, protein, lipid, and other small molecules) produced by all types of living cells [Figure 2]. Exosomes represent a smaller type of EVs released from the cells through the exocytosis of multivesicular bodies with a range of size from 50 to 150 nm [Figure 2]. Another type of EVs are the microvesicles (large EVs) or ectosomes which arise through direct budding from the cell membrane. They range in size from 100 to 1000 nm [118]. Cell-free therapy using EVs offers several advantages over treatment using MSCs by themselves. EVs have no risk of aneuploidy or immune reactions, which may be a complication of cellular therapy. The EVs, secreted by MSCs (MSCs-EVs), contain a range of biomolecules, including proteins, lipids, and nucleic acids, closely mirroring the macromolecular profile of parent MSCs. Consequently, MSCs-EVs have emerged as promising therapeutic agents for various disorders, highlighting the pivotal role of paracrine effects in MSCs-based therapies. We showed that placenta and umbilical cord blood-derived exosomes (small EVs) play an important role in tissue regeneration, such as the expansion of hematopoietic stem cells [55,119].

MSCs were initially believed to exert their therapeutic effects through direct differentiation or engraftment at the site of tissue injury, but research has shown that less than 1% of transplanted MSCs reach the target tissue, with most getting trapped in organs such as the liver, spleen, and lung. Instead, mounting evidence supports the idea that MSCs primarily influence resident cells through the secretion of paracrine bioactive components. These components may be either directly released to the surroundings as soluble factors such as cytokines and growth factors or contained in vesicles to be shed by the cells as extracellular vesicles (EVs) [120].

### 10.1. Role of MSCs Derived Extracellular Vesicles (EVs) in TL Shortening, Cellular Senescence, and Biological Age 

EVs are associated with several hallmarks of aging, including TL. Stem cell-derived EVs are considered the main effectors of intercellular communication between MSCs and cells of target tissues transferring biomolecules that play a key role in the aggravation or mitigation of the hallmarks of aging [121].

Some recent studies showed the possible role of EVs in maintaining TL in the recipient cells through various mechanisms. These mechanisms include transferring proteins, mRNAs, or even the direct transfer of telomeric DNA elements from one cell to another.

EVs carry proteins that are related to telomere maintenance and function. For example, recent studies showed that EVs from amniocytes and cancer cells carry the hTERT protein [122,123]. These EVs were able to increase the life span of the recipient cells, indicating the possible role of EVs in maintaining telomere length in the recipient cells by transferring hTERT protein [123]. Another study suggested the transfer of proliferating cell nuclear antigen (PCNA) mRNA as a mechanism for the maintenance of TL induced by UCMSCs. These UCMSCs-EVs rejuvenated the adult bone marrow MSCs (ABMSCs), as indicated by reduced senescence and increased TL. According to their results, the transfer of PCNA mRNA enabled the expression of PCNA protein in the recipient cells, which maintained TL and participated actively in alleviating senescence [124]. PCNA could maintain the TL through the alternative lengthening of telomeres (ALTs) pathway [125]. In another context, small EVs were found to directly transfer telomeric DNA from antigen-presenting cells to T cells, contributing to T-cell telomere elongation with subsequent extended proliferation capacity, which is essential for the formation of memory cells [126].

The role of EVs as SASP factors is largely offset by their potential as senotherapeutics. Young MSC-derived EVs rescued the function of aged stem cells and represent an effective and safe approach for conferring therapeutic effects of stem cells and slowing the progression of aging and diseases driven by cellular senescence [127]. When observing the differences between young MSCs-EVs and aged MSCs, young MSCs-EVs have greater anti-inflammatory activity [120].

### 10.2. Antiaging Effect of MSCs-Derived EVs

EVs, as one of the fundamental mediators of intercellular communication, might play a key role in the aggravation or mitigation of the hallmarks of aging and may provide context for the future implementation of multiple emerging EV-based therapeutic strategies that are currently under study [121] EVs derived from different types of stem cells show antiaging and antisenescence properties. They also have been reported to control TL in the recipient cells. For example, embryonic stem (ES) cell-derived EVs (ES-EVs) possess antisenescence properties, as they significantly alleviated the senescence that occurred in the late-passage placental MSCs (passage 18). Additionally, the ES-EVs improved the wound-healing capacity of these MSCs in vivo. This approach could potentially emerge as an innovative therapeutic strategy for the clinical application of MSCs. [128]. Other stem cells, such as human-induced pluripotent stem cells (iPSCs), appear to have therapeutic effects and can prevent UV-B cell injury through their small EVs or exosomes. This was demonstrated through the pretreatment of human dermal fibroblasts (HDSs) with iPSC exosomes. Exosomes stimulated the proliferation and migration of dermal fibroblasts and reduced senescence markers and matrix-degrading enzymes caused by the UVB radiation, and they were associated with the prevention of cell-induced injury from UVB. This was thought to be achieved by increasing the expression of SA β-GAL and MMP (1/3), as well as restoring the collagen type 1 expression in senescent HDFs [129]. As a result, there is therapeutic potential for those HiPSC-derived exosomes in treating skin aging. In summary, EVs from different cell sources regulate cell longevity through the regulation of TL. Different mechanisms are suggested, including the transfer of telomerase proteins, the transfer of mRNA of proteins that maintain telomere with the ALT pathway, or the direct transfer of telomere DNA. Further studies are required to understand these different mechanisms fully and explain how these EVs can be used to reduce cellular senescence and prevent TL shortening and the aging process.

## 11. Conclusions

This review serves as an overview of the factors that may lead to TS and accelerated biological aging, the effects of inflammatory and oxidative processes that can lead to cellular senescence and accelerated biological aging. The review discusses the mechanisms and the advantages of MSCs and, more specifically, gestational tissue-derived MSCs in preventing TS, cellular senescence, and accelerated biological aging. The review also discusses the role of MSCS-derived EVs in the prevention of TL shortening, cellular senescence, and accelerated biological aging. The antiaging therapeutic applications of MSCS and EVs were reviewed, as well. More research is needed to expand our knowledge in this novel area of stem cell therapy and cell-free therapy and their role in haltering or reversing TS and cellular senescence. This review can serve as an introduction to exploring the role of stem cell therapy, including MSCs and their derived EVs in preventing cellular senescence and their potential therapeutic application against age-related diseases.

## Figures and Tables

**Figure 1 bioengineering-11-00524-f001:**
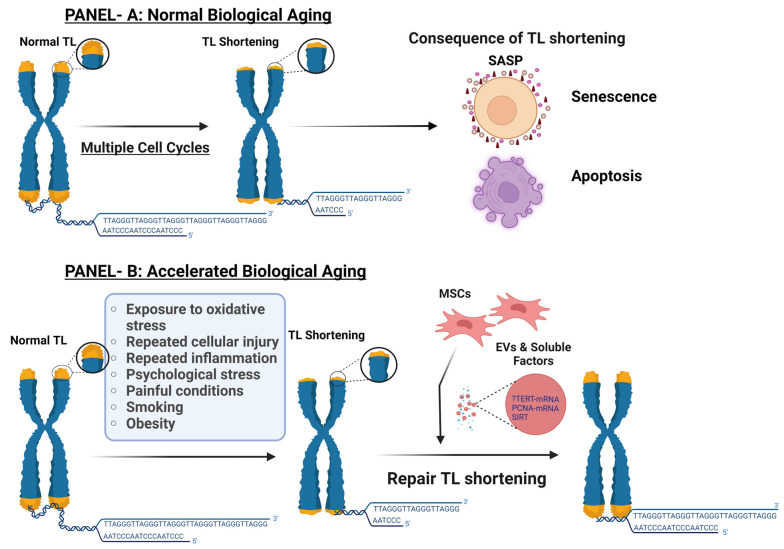
This illustration summarizes the normal and accelerated biological aging and associated shortening of telomere length (TL). Panel A depicts Telomere length shortening (TS) as a natural process that occurs after each cell division, progressing at a steady rate throughout the aging process. Panel B depicts accelerated biological aging associated with TS. This can occur under specific abnormal conditions, as outlined in the figure. This illustration was created using BioRender.com.

**Figure 2 bioengineering-11-00524-f002:**
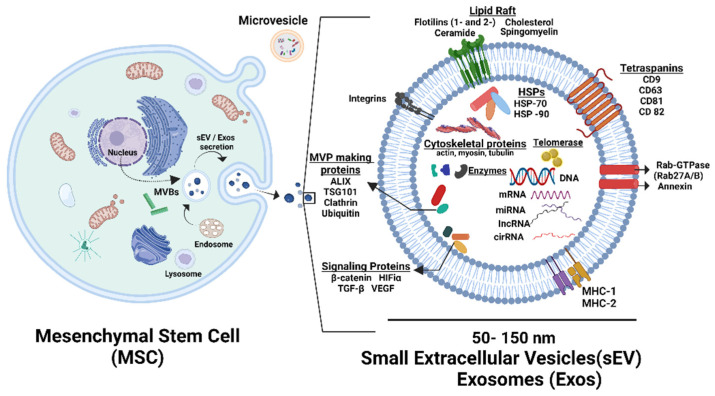
Schematic diagram illustration of small EVs (sEVs) or exosomes derived from MSC. Small EV is a 50–150 nm lipid bilayer structure with surface markers and contains biomolecules, including proteins; lipids; and nucleic acids, such as mRNA and miRNA. This illustration was created using BioRender.com.

## Data Availability

All data and material used for writing the manuscript are available.

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
