# Peer review of "Role of Mesenchymal Stem/Stromal Cells (MSCs) and MSC-Derived Extracellular Vesicles (EVs) in Prevention of Telomere Length Shortening, Cellular Senescence, and Accelerated Biological Aging"

_bioengineering, 2024, doi:10.3390/bioengineering11060524_

Round 1

Reviewer 1 Report

Comments and Suggestions for Authors

The review is of interest, particularly the topic of ageing. The figures are of good quality,

It is recommended that the authors include a section about the role of aged MSCs in some diseases in which MSCs are playing a role, and their ageing is directly contributing to tissue damage or inflammation.

Also, highlighting the importance of this reserch for therapeutic applications is needed. 

There is a particular focus on placental MSCs, but since the review is about MSCs, more about other types, such as bone marrow MSCs that reside in skeletal tissues, should be included.

Comments on the Quality of English Language

Overall, a good quality of English, but some minor grammar and typos can be corrected.

Author Response

Response to the reviewers’ comments

The authors  appreciate the comment by the reviewers. Below are the authors’ response to the reviewers’ comments in bold as reflected in the submitted revised manuscript.

Reviewer 1 comments:

Comments and Suggestions for Authors

The review is of interest, particularly the topic of ageing. The figures are of good quality,

It is recommended that the authors include a section about the role of aged MSCs in some diseases in which MSCs are playing a role, and their ageing is directly contributing to tissue damage or inflammation.

We appreciate the comment by the reviewer. The authors agree with the reviewer suggested changes.

Section 6. Role of aged MSCs in pathogenesis of diseases was added to address the role of aged MSCs in some diseases in which MSCs are playing a role, and their ageing is directly contributing to tissue damage or inflammation.

Also, highlighting the importance of this research for therapeutic applications is needed.

Section 6. Antiaging therapeutic application of MSCs was added to address the reviewer comment.

There is a particular focus on placental MSCs, but since the review is about MSCs, more about other types, such as bone marrow MSCs that reside in skeletal tissues, should be included.

Section 4.1.1. Bone Marrow MSCs was added to address the reviewer comment.

English Language editing for the whole manuscript was completed.

Reviewer 2 Report

Comments and Suggestions for Authors

Review article cover wide spectrum of events related to MSC-based therapy and ageing.  The main question addressed by this article - to summarize current understanding of contribution of MSCs and EC from MSC to telomere length and ageing.   The topic of article is addressed to new topic (MSC-based therapy and ageing via effects on telomere length) but mechanisms were not described precisely.  In comparison to other published material  new direction of research was traced. And conclusions are consistent with the arguments presented. References are appropriated to described materials.                             Therefore it is recommended to made focus of telomerase activity and telomere length in senescence, stress reactions, inflammation, oncogenesis. If authors discuss ageing there are many mechanisms to be targeted by MSCs that should be described. Regarding comparison of human and mouse MSCs - there are some differences between them including their capability to be transformed to tumor cell type. It should be also added that MSC- based therapy is based on short - lived mesenchymal stromal cells that  provide biological effect predominantly not contact by in paracrine manner. So additional mechanisms should be described or proposed for down-stream events related to telomere length and ageing.

In conclusion review article should be shortened for events to be precisely describes including telomerase/telomere length, MSCs and EV of MSCs, ageing

Author Response

Response to the reviewers’ comments

The authors  appreciate the comment by the reviewers. Below are the authors’ response to the reviewers’ comments in bold as reflected in the submitted revised manuscript.

Reviewer 2 Comments:

Comments and Suggestions for Authors

Review article cover wide spectrum of events related to MSC-based therapy and ageing.  The main question addressed by this article - to summarize current understanding of contribution of MSCs and EC from MSC to telomere length and ageing.   The topic of article is addressed to new topic (MSC-based therapy and ageing via effects on telomere length) but mechanisms were not described precisely.

To address the reviewer comment, the mechanisms were described in

Section 5.2. Mechanisms of MSCs cellular senescence

In comparison to other published material  new direction of research was traced. And conclusions are consistent with the arguments presented. References are appropriated to described materials.                            

Therefore it is recommended to made focus of telomerase activity and telomere length in senescence, stress reactions, inflammation, oncogenesis.

To address the reviewer comment, please see

  • Section 5.3. Role of MSCs in Telomere length and Biological Age
  • 2. Mechanisms of MSCs cellular senescence
  • Subsection 5.2.6. Oncogene-induced cellular senescence

If authors discuss ageing there are many mechanisms to be targeted by MSCs that should be described. Regarding comparison of human and mouse MSCs - there are some differences between them including their capability to be transformed to tumor cell type.

To address the reviewer comments, the following sections were added,

  • 2. Mechanisms of MSCs cellular senescence
  • Subsection 5.2.6. Oncogene-induced cellular senescence
  • Section 5.4. Strategies to prevent MSCs Senescence
  • Section 6. Antiaging therapeutic application of MSCs

It should be also added that MSC- based therapy is based on short - lived mesenchymal stromal cells that  provide biological effect predominantly not contact by in paracrine manner.

Section 8.1. Paracrine Effect of MSCs by Extracellular vesicles (EVs) was added to address the reviewer’s comment.

So additional mechanisms should be described or proposed for down-stream events related to telomere length and ageing.

To address the reviewer comment, the following section and subsections were added:

Section 5.2. (5.2.1 to 5.2.6) Mechanisms of MSCs cellular senescence

And

Section 5.4. Strategies to prevent MSCs Senescence

In conclusion review article should be shortened for events to be precisely describes including telomerase/telomere length, MSCs and EV of MSCs, ageing

The manuscript was reviewed thoroughly and any possible section that can be shortened was removed.

Round 2

Reviewer 2 Report

Comments and Suggestions for Authors

Current version of Review has new facts and necessary comments regarding MSCs participation  in anti-aging strategy. Involvement of telomerase-depending mechanisms, oxidative stress now is described more detailed. Paracrine mechanisms of MSCs action now is described and discussed more widely.

The purpose of review to prove MSCs anti-aging effects is proved more effectively.

Author Response

The reviewer accepted our revision. Attached response to the  editor requested changes.
